# Microbial community succession mediated by planting patterns in the Loess Plateau, China: Implications for ecological restoration

Weiqian Li[1,2], Jinjun Cai●[1,2*], Gang Chen[1,2], Yitong Liu[1,2], Xia Wu[1,2], Yangyang Bai[1,2], Yan Wu[1,2], Tianning Wang[1,2]

**1** Institute of Agricultural Resources and Environment, Ningxia Academy of Agriculture and Forestry Sciences, Yinchuan, P.R. China, **2** The Key Laboratory of Soil and Plant Nutrition of Ningxia, Yinchuan, P.R. China.

\* nxyccai@163.com

## Abstract

Microbial community succession plays a key role in restoring fragile ecosystems and mitigating ecological degradation. However, the mechanisms by which vegetation restoration promotes ecological restoration and microbial community reconstruction in degraded soils remain unclear. This study utilized metagenomic high-throughput sequencing technology to analyze microbial community dynamics in soil samples collected from eight different planting patterns in the ecologically degraded areas of the Chinese Loess Plateau. The results indicated significant effects of terrain location and restorative cropping patterns on soil microbial abundance and function. In particular, soil C and N nutrient abundance was highest in mixed forest soils, and the total number of microorganisms was highest and more diverse. Therefore, through vegetation restoration, mixed forests significantly enhanced regional ecological functions. Notably, creating mixed forests with both trees and shrubs resulted in optimal ecological functions, providing a valuable direction for vegetation construction and structural optimization in the region.

## 1. Introduction

The loess beam-shaped hilly area is an important part of the Loess Plateau in China. The Loess Plateau has undergone significant degradation due to human activities [1], necessitating large-scale restoration efforts such as the Grain-for-Green Program (GFGP) [2,3]. However, while afforestation efforts have improved vegetation cover, their effects on microbial succession and soil health remain unclear. The arbitrary selection of tree species without considering microbial interactions may contribute to unintended ecosystem shifts [4,5]. Searching for a suitable regenerative afforestation model may be key to solving this problem.

**Data availability statement:** All relevant data are within the paper and its Supporting Information files.

**Funding:** The authors wish to express their sincere thanks to Natural Science Foundation of Ningxia Hui Autonomous Region (2024AAC03372), the Ningxia Hui Autonomous Region Science and Technology Innovation Leading Talent Project (2023GKLRLX20) and the Ningxia Hui Autonomous Region Key R&D Project (2023BEG02042). In addition, this work also is supported by the special project of Ningxia Academy of Agriculture and Forestry Science with foreign (DW-X-2020002), the Ningxia Hui Autonomous Region Sci-Tech innovation demonstration program of High-quality agricultural development and ecological conservation (NGSB-2021-11-01), the Special Projects for The Central Government to Guide the Development of Local Science and Technology (2021FRD05023). The authors thank Biomarker Technologies for the support of tests.

**Competing interests:** The authors have declared that no competing interests exist.

Different microorganisms with distinct functional characteristics in the soil are essential and active ecosystem components. They play a key role as the main drivers of soil nutrient cycling and serve as important indicators of soil quality and productivity [6]. Despite comprising less than 0.5% of soil mass, microorganisms drive critical biochemical processes essential for soil fertility and ecosystem stability [7]. Soil microorganisms play a vital role in the cycling of mineral elements, the formation and decomposition of organic matter, the breakdown of sedimentary materials in the natural environment, plant growth and development, and the prevention and management of crop diseases [8]. Previous studies have shown that soil microbial activity and community structure change significantly under different types of land use. Vegetation in restored areas generally includes grassland, forest, and agroforestry zones. Grassland vegetation, with its shallow root systems, has a relatively limited effect on soil layers and associated microbial communities compared to forests, where deeper root systems affect a wider range of soil depths [9]. Furthermore, mixed forests with nitrogen-fixing tree species have been shown to restore soil quality and ecological functions better than monoculture plantations. In temperate artificial forests located on the heterogeneous deep loess layers, microbial communities are likely to have distinct compositions along soil profiles, thereby influencing ecosystem feedback [10]. These findings highlight that geographic location, environmental conditions, and vegetation type are critical factors shaping the structure, activity, and diversity of microbial communities. Successional changes in vegetation significantly alter microbial abundance, activity, community structure, functionality, and diversity [11]. However, most current research on soil microbial ecosystems focuses on specific species, habitats, forest types, or soil depths, including forest soils, degraded grasslands, nature reserves, orchards, and mining areas. In contrast, studies on microbial structural composition and diversity driven by vegetation differences within small-scale areas remain limited.

Metagenomics is of great significance in ecology, especially in practical research on the community composition and structure of soil microorganisms. Traditionally, the analysis of microbial diversity is based on a separation method and further characterization of the obtained colonies, which ignores the unculturable portion of the microbial population [12]. This is especially true for microbes that exist in complex microbial ecosystems. Next-generation sequencing (NGS) technology can better characterize microbial diversity in complex ecosystems [13]. Priya et al. [14] suggested that genomics is a potential tool for unraveling the interactions between the rhizosphere microbiome and plant ecosystem. It is believed that plants-soil-microorganisms have a "co-inheritance" effect in the symbiotic system of plants, soil, and microorganisms, with similar spatial differentiation and distribution on different scales [15]. Hu et al. [16] emphasized that the loss of plant and soil microbial biodiversity may have serious consequences under low and high drought conditions. Therefore, we propose the establishment of biodiversity conservation strategies for various planting patterns in areas with degraded soil to reduce the impact of soil degradation. Moreover, the study showed that vegetation communities showed strong distribution characteristics under human intervention. However,

whether the complexity of the plant community results in a distribution pattern similar to that of the structure and composition of the plant rhizosphere or even the soil microbial community is still unclear. It is known that plant-soil feedback can promote coordinated adaptation and may, therefore, select factors that increase the resistance of degraded ecosystems and improve the beneficial association with soil organisms [17]. Negative plant-soil feedback may be transformed into a positive effect of the soil microbial community on plant growth over time [18]. Although previous research has explored vegetation restoration on degraded soils, the specific mechanisms through which different planting patterns affect microbial community succession and soil function remain unclear. Therefore, we hypothesize that mixed planting systems will enhance microbial diversity, increase functional gene expression, and improve soil enzyme activity more effectively than monoculture systems. Based on second-generation NGS technology, this study investigates the response mechanisms of microbial community composition and structural changes to eight different types of vegetation restoration in the degraded areas of the Loess Plateau, to improve the understanding of soil microorganisms and their feedback on the ecological environment and functions, thereby provide scientific evidence for the region.

## 2. Materials and methods

### 2.1. Study area

This research was conducted in the Zhongzhuang Village Forest Farm in Baiyang Township, Pengyang County, Guyuan City, Ningxia Hui Autonomous Region, China (latitude 35°51′–35°55′ north, longitude 106°45′–106°18′ east). A detailed map of the study area is shown in Fig 1. The landform type of the study area was the Liangmao hilly land in the central region of the Loess Plateau, with a total area of 65,290 square kilometers. The annual temperature in this area is 7.63 °C,

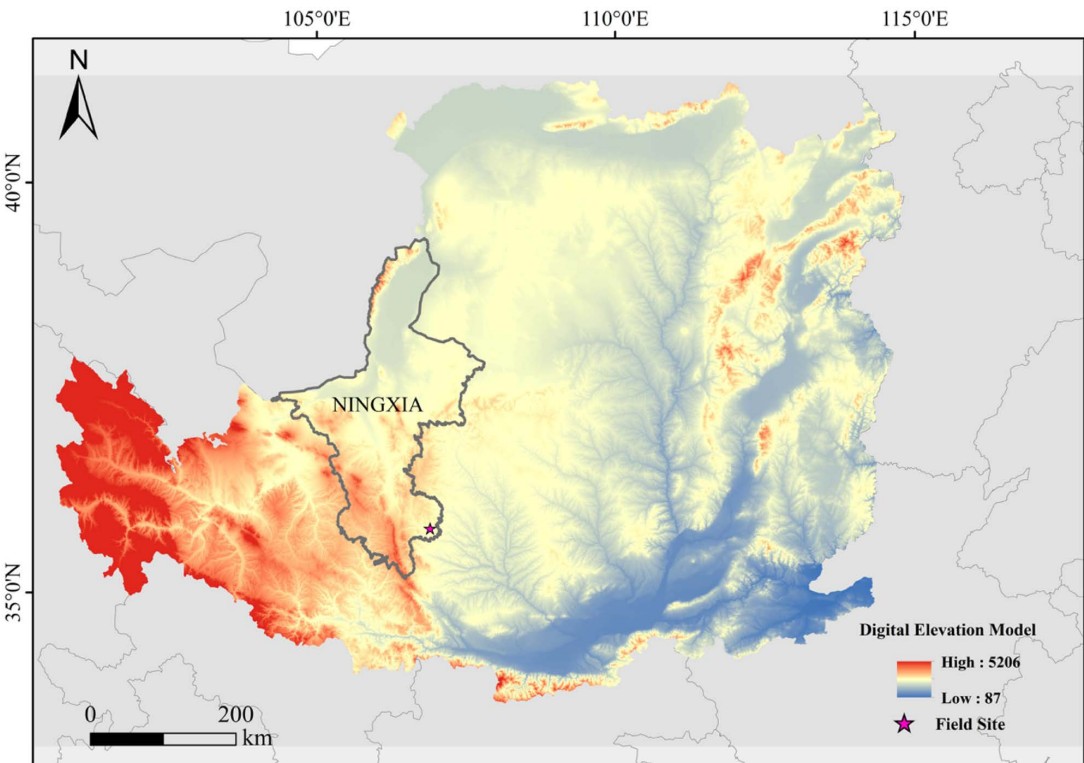

**Fig 1. Location map of the study area (Resources by Shuttle Radar Topography Mission Digital Elevation Model -90 meters resolution elevation data.** http://www.gscloud.cn).

the accumulated temperature of ≥ 10°C is 22.00–27.50 °C, the annual evaporation in the territory is relatively high, the dryness (evaporation ≥ 0°C) is 1.21–1.99, the frost-free period is 140–160 days, the vegetation cover rate was 11%, and the annual precipitation was 483 mm.

Eight land cultivation areas with similar geomorphological conditions were selected for sampling. The selection method for these sites was based on planting patterns. The names and abbreviations of the sampling sites are listed in Table 1. In addition, the temporal change was usually lower than the spatial change in a few cases where it has been clearly quantified. This is because in forests, although temporal changes can affect microbial community structure, these effects typically manifest themselves as seasonal variations or long-term trends. In contrast, spatial variation is more immediate and pronounced because environmental factors tend to exhibit greater complexity and diversity in spatial heterogeneity than in temporal changes. Therefore, this study combined all samples from each group at a uniform sampling time and analyzed the soil physics and chemistry, soil enzyme activity, and soil microbiome after collection.

## 2.2. Soil sampling and sample preparation

Surface soil samples were collected from the plant cultivation areas at eight selected sites. Soil microorganisms are primarily concentrated in the surface layers because these soils are most directly exposed to the external environment, which promotes microbial survival and proliferation. In addition, surface soils are rich in organic matter and moisture, providing favorable living conditions for microorganisms. In forests, plant roots are more concentrated in the 0-0.20m depth range, and root exudates provide abundant carbon and nitrogen sources for microorganisms, facilitating their growth and reproduction. While deeper soil layers may also harbor microbial communities, the 0–0.20 m depth was chosen as it is most influenced by plant root activity and soil disturbance. Therefore, soil samples were collected at a depth of 0-0.20m. Representative plant species were selected based on dominance, ecological importance, and their role in the restoration process. Different vegetation configurations, including pure forest, forest and shrub, forest-shrub-grass, pure grass, and abandonment were selected as research objects in the Zhongzhuang watershed in July 2020. The vegetation types of returning farmland to forests, including AR, AM, AR&HR, CA&AR, AM&AR&MS, MS, AL, and other vegetation types were selected as test plots. The woodlands were mature forests and abandoned uncultivated land was used as the control plot. Three sample plots (20 m × 20 m) were randomly set for each vegetation type, and three small plots (1.0 m × 1.0 m) were set in the plots. Three replicates per vegetation type were used to ensure statistical robustness and account for site-level variability. Soil samples were collected using a metal soil core sampler (15 cm inner diameter). The top 0.20 m depth was chosen because it represents the soil layer most disturbed by tillage and erosion [19]. A part of the sample was passed through a 2 mm sieve for soil physical, chemical, and enzymatic properties, and the remaining part was retained at a rapid low temperature (liquid nitrogen quick freezing) without sieving for microbial aggregate analysis.

**Table 1. Sampling point naming and abbreviations.**

| Point of sampling | abbreviations |
|---|---|
| Abandoned land | AL |
| *Medicago sativa* L. (Alfalfa) grassland | MS |
| *Armeniaca sibirica* forest | AR |
| *Amygdalus davidiana* forest | AM |
| *Armeniaca sibirica - Hippophae rhamnoides* Mixed Forest | AR&HR |
| *Armeniaca sibirica - Caragana* mixed forest | CA&AR |
| *Armeniaca sibirica - Amygdalus davidiana-Alfalfa* mixed forest | AM&AR&MS |

## 2.3. Determination of soil physicochemical and enzymatic properties

The soil moisture content was determined by the drying method, and the soil bulk density was determined by sampling and weighing with a ring knife. A glass electrode was used to determine the pH of the soil. Soil organic carbon (SOC) and total nitrogen (TN) were determined using standard procedures of dichromate oxidation and Kjeldahl nitrogen digestion, respectively [20]. Total phosphorus (TP) and available phosphorus were extracted with $H_2SO_4$-$HClO_4$ and sodium bicarbonate, respectively, and then measured by the molybdenum blue method at 760 nm with an ultraviolet spectrophotometer. Sucrase and cellulase activity was measured by the 3,5-dinitrosalicylic acid colorimetric method; urease measured by sodium phenate-sodium hypochlorite colorimetric method [21]; phosphatase activity measured by phenyl disodium phosphate colorimetric method; catalase activity measured by Potassium permanganate titration method; peroxidase and polyphenol oxidase measured by iodometric titration method. These enzymes were selected as indicators of soil microbial activity and nutrient cycling, which are essential for assessing ecological restoration progress. All the indicators were measured by spectrophotometer (V5100, Metash, Shanghai, China).

## 2.4. Soil DNA extraction, amplification, and Illumina MiSeq high-throughput sequencing

**2.4.1. Extraction of DNA from soil samples.** Herein, the method described by Fierer et al. was used to quantify the relative abundance of bacterial and fungal rRNA gene copies [22]. Power Soil® DNA kit (MOBIO Laboratories, Carlsbad, CA, USA) to extract soil microbial DNA, 1% agarose gel to detect DNA, Gel recovery kit (TianGen Biotech Co. Ltd) to purify DNA, and then set it after purification. All soil samples were stored in a refrigerator at -40°C, each soil sample has 3 replicates. Otherwise, the Power Soil® DNA kit has been widely used in DNA extraction and is considered adequate to meet the needs of this study.

**2.4.2. qRT-PCR quantitative analysis.** The PCR process using TransStartFastPfu DNA polymerase, and the PCR instrument by ABI GeneAmp®9700. 20 μL PCR reaction system: 5×FastPfu buffer, 4 μL; 2.5 mmol·L-1 nucleotide, 2 μL; forward primer (5μmol·L-1), 0.8 μL; reverse primer (5μmol·L-1), 0.8 μL; 0.4 μL FastPfu polymerase; template DNA, 10 ng; added $H_2O$ to 20 μL. PCR reaction steps and conditions: denaturation at 95°C for 30 s to break the hydrogen bond of the DNA double helix to form single-stranded DNA as a reaction template; annealing at 55°C for 30 s, the primer and the complementary region of the template combine to form a template-primer complex; extension at 72°C for 45 s, using the primer as a fixed starting point to synthesize a new DNA strand under the action of DNA polymerase. The above three steps were repeated as a cycle, a total of 30 cycles. The PCR products were quantified by QuantiFluor TM-ST blue fluorescence system, then mixed according to the sequencing amount of each sample, and finally the MiSeq library was constructed and sorted.

**2.4.3. Illumina MiSeq high-throughput sequencing.** We used the FastQC software to check data quality and generated a web version of the quality control report and data FASTQC.zip. Then use Fastp for low-quality data filtering. The original sequence was cut, merged, and distributed in the QIIME software according to the standards (v1.8.0, http://www.qiime.org/). Remove low-quality sequences (length <150 bp). High-quality sequences were clustered into operational taxa (OTUs) with 97% similarity. Calculate the number of OTUs, β diversity index, Chao1 index, Shannon, and Simpson indexes in QIIME. Meanwhile, the ANOSIM analysis method was used to analyze the differences between the treatments.

## 2.5. Statistical analysis

Without special note, all analyses were performed at least in triplicate. Analysis of variance was used to determine the significance of differences in microbial diversity index (Shannon), abundance, and physical and chemical characteristics of soil and litter between succession stages. A significant difference was determined at the 95% confidence level. When significance was detected at the $P < 0.05$ level, the post-hoc Duncan multiple range test was used for multiple comparisons. The Welch's t-test is more robust and applicable when comparing differences in sample means. Therefore, Welch's t-test

was used to analyze the differences between the two sample groups. Welch's t-test, or t' test, was used to test the difference between two sets of samples. ANOVA (Analysis of Variance, namely: analysis of variance) was also known as F test (F test). Before statistical tests are performed, a significance level (e.g., 0.05, 0.01, etc.) is preset to determine whether the difference is significant. This significance level is a threshold used to determine whether the observed difference is sufficient to reject the null hypothesis (i.e., the hypothesis that there is no significant difference between the two groups). The assembly of metagenomics used the software MEGAHIT [23] to assemble the metagenomics, filtering contig sequences shorter than 300 bp. Using QUAST [24] software to evaluate the assembly results. Performing functional annotations on the biological information obtained, including general database annotations such as Kyoto Encyclopedia of Genes and Genomes (KEGG), Evolutionary Genealogy of Genes: Non-supervised Orthologous Groups (eggNOG), the protein family database (Pfam, MSA), SwissProt, Non-redundant protein sequence database (NR), Gene Ontology (GO), and special database annotations such as Comprehensive Antibiotic Resistance Database (CARD), Carbohydrate-Active enZYmes Database (CAZy); and comparing and analyzing the functional diversity of each sample, including functional Principal component analysis (PCA) Analysis, functional PCoA analysis, Network network analysis, anosim group similarity analysis, and differential functional gene analysis. MetagenomeSeq was chosen due to its effectiveness in addressing sparse microbiome datasets, reducing bias in differential abundance analysis.

## 3. Results and discussion

### 3.1. Soil physicochemical and biological enzymatic activities under different planting patterns

Chemical, physical, and biological properties are widely recognized as essential indicators of soil quality [25]. Soil microorganisms are an important component of genetic diversity in terrestrial ecosystems and play an important role in the subsurface carbon (C) cycle of terrestrial ecosystems. Active nitrogen enters the terrestrial ecosystem through atmospheric sedimentation, which can effectively alleviate the nitrogen limitation of above-ground vegetation, promote plant growth, and increase plant biomass. However, too much nitrogen may cause soil pH to decrease, resulting in soil nutrient imbalance and damage to the stability of the ecosystem. In the process of soil carbon cycling, soil microbial diversity and soil organic carbon were closely coupled. This coupling could be weakened or broken by rapid global nitrogen deposition. Soil stability and soil aggregates, which are highly susceptible to land degradation, are important drivers of soil fertility and microbial diversity, which are highly susceptible to land degradation [26]. Most soil parameters differed significantly between the rhizosphere and bulk soil among the planting types (**Table 2**). Compared with the single planting mode, the total carbon (TC) and total organic carbon (TOC) contents in soil under the multi-plant planting mode increased, which was attributed to the input of plant biomass carbon [27]. Soil organic carbon (SOC) content is correlated with microbial abundance because SOC provides substrates for microbial metabolism [28]. The increase in SOC content in the multi-plant vegetation restoration model was conducive to the formation of abundant soil microorganisms in the woodland soil.

Table 2. Soil physicochemical under different planting patterns.

| Sample ID | pH | EC (Ms/cm) | C[Na⁺] (mg/kg) | C[K⁺] (mg/kg) | TC (mg/g) | TOC (mg/g) |
|---|---|---|---|---|---|---|
| AL | 8.27 | 0.13 | 0.71 | 0.72 | 0.87 | 0.63 |
| CA&AR | 8.25 | 0.17 | 0.94 | 0.68 | 1.55 | 1.38 |
| MS | 8.23 | 0.16 | 0.86 | 0.53 | 1.25 | 1.01 |
| AM&AR&MS | 8.33 | 0.12 | 0.59 | 0.97 | 1.89 | 1.38 |
| AM | 8.23 | 0.12 | 0.62 | 0.92 | 1.73 | 1.35 |
| CA | 8.18 | 0.14 | 0.45 | 0.41 | 1.42 | 1.13 |
| AR | 8.22 | 0.11 | 0.68 | 0.65 | 0.83 | 0.68 |
| AR&HR | 8.21 | 0.10 | 0.44 | 0.51 | 1.57 | 0.92 |

Statistical analysis showed that the catalase content in the planting area did not change significantly. However, phosphatase, sucrase, BG, and NAG levels were significantly lower in the recovered planting area than in the abandoned area ($p < 0.05$; Fig 2). Higher phosphatase and sucrase activities indicate enhanced microbial nutrient cycling, which is essential for soil restoration and plant nutrient availability. The increased abundance of sucrase and phosphatase activity in mixed planting areas suggests enhanced microbial decomposition of organic matter and phosphorus mobilization, contributing to soil restoration. This indicates that the plants used for restorative planting should be properly matched to form a more suitable restored forestland [29]. $NO_3^-$-N fluctuated greatly with changes in planting type. Compared with abandoned land, the content of $NH_4^+$-N and TN increased with the removal of wilderness ($p < 0.05$, Fig 3). The observed increase in $NH_4^+$-N and TN under restorative planting suggests that plant-soil interactions enhance nitrogen retention, while fluctuating $NO_3^-$-N levels may indicate differential microbial uptake or leaching patterns. In general, the content of soil nitrogen species increased with increasing restorative planting diversity, indicating that restorative planting with multiple plants was more suitable for soil fertility recovery. Soil nutrient element ratios reflect the coupling of carbon (C), nitrogen (N), phosphorus (P), and other elements in terrestrial ecosystems and can be used to explore how microorganisms respond and adapt to changing environments [30]. Soil C, N, and P concentrations provide the basic energy and nutrient requirements for soil microorganisms. Soil C and N can regulate the composition of the microbial community and maintain the balance between element absorption and release in the plant-soil system [31]. As the multi-plant restoration model progressed, the contents of soil nitrogen, soil carbon, and related species increased, indicating that the multi-plant model was more conducive to the restoration of soil ecology. However, an appropriate mixture of plant types is required to ensure optimal performance.

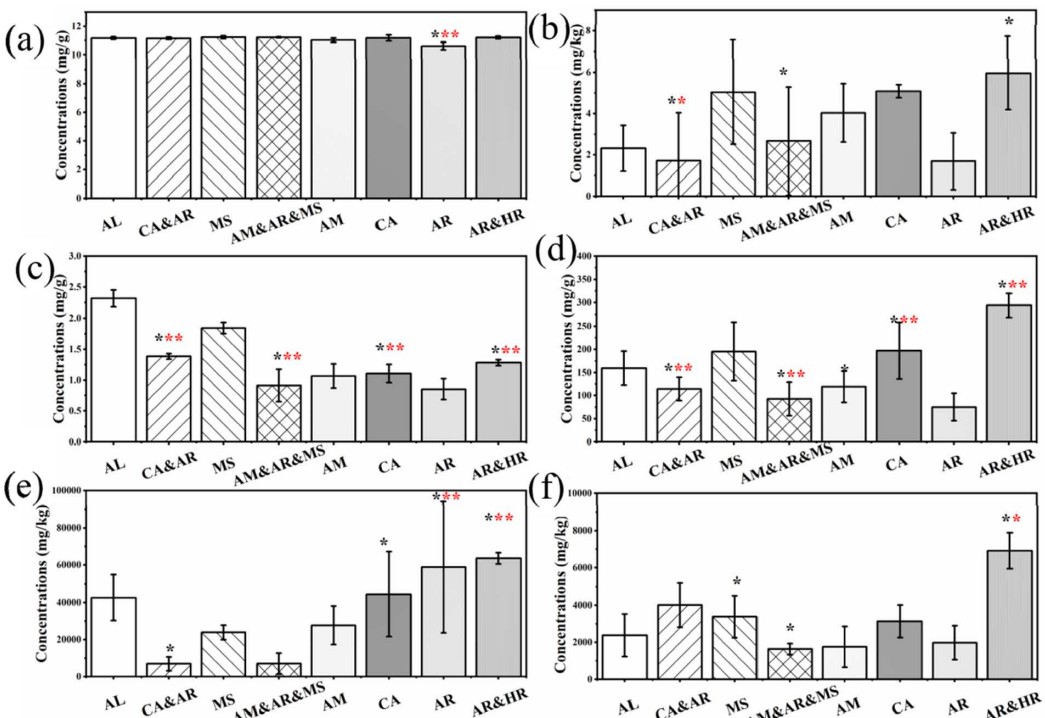

**Fig 2. Soil enzyme activity.** a, catalase content; b, urease; c, phosphatase; d, sucrase; e, β-glucosidase (BG); f, β-N-Acetamido-glucosidase (NAG).

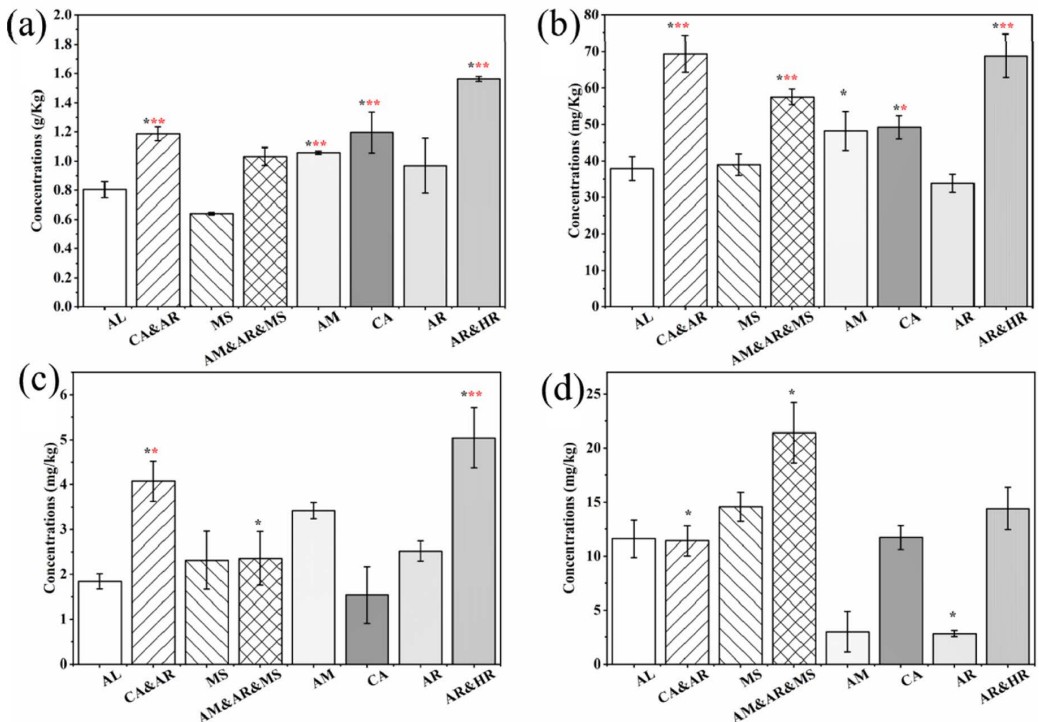

**Fig 3. Soil ammonia nitrogen.** a, total nitrogen; b, hydrolyzable nitrogen; c, ammonia nitrogen; d, nitrate nitrogen.

### 3.2. Soil microbiological composition under different planting patterns

Classification and identification of sequencing results showed that the main composition of microorganisms was bacteria, accounting for 83.2–83.9% of all microbial groups; fungi, archaea, and viruses only exist for 1.2–2.3% (Table A1, Fig 4a and 4b). The main detected bacteria were *Acidobacteria*, *Gemmatimonadetas*, *Candidatus*, *Verrucomicrobia*, *Sphingomonas*, *Chloroflexi*, and *Betaproteobacteria* (Fig 4c and 4d); *Acidobacteria* had the highest abundance (17.6–18.3%). It indicated that as far as the dominant bacteria were concerned, the same dominant bacteria, *Acidobacteria*, were detected at each sampling point. *Acidobacteria* are among the most important bacterial groups in soils worldwide [32]. The dominance of *Acidobacteria* in the microbial community is ecologically significant, particularly in nutrient-poor and acidic soils, where these bacteria thrive. Acidobacteria contribute to soil processes such as nutrient cycling, organic matter decomposition, and pH regulation, all of which are crucial for soil health and restoration [33]. In some cases, it accounted for 52% of the total bacterial community. Changes in soil ecological restoration from derelict to artificial grasslands and woodlands can disrupt and affect the soil environment and communities significantly. Changes in biodiversity occurred, and the complex restoration model promoted soil biodiversity and increased the complexity and stability of soil micro-community structure. However, studies have shown that *Acidobacteria's* succession and community dynamics are very poor, especially after artificial planting pattern changes. The acidic bacterial community is affected by soil pH, pesticides, heavy metals (Zn, Cr, Ni, Cu, Cd, and As), and terminal electron acceptors ($NO_3^-$, bioavailable Fe (III), and $SO_4^{2-}$). This is believed to be related to the long history of artificial fertilization [34]. Changes in the planting type led to a downward trend in the relative abundance of *Acidobacteria*. This change from mixed forests to single crops led to changes in key soil chemical characteristics, such as pH, distribution of artificial ammonia nitrogen, and enzyme activity, which subsequently affected the bacterial community structure [35]. Beyond *Acidobacteria*, *Gemmatimonadetes* and *Verrucomicrobia* also play crucial

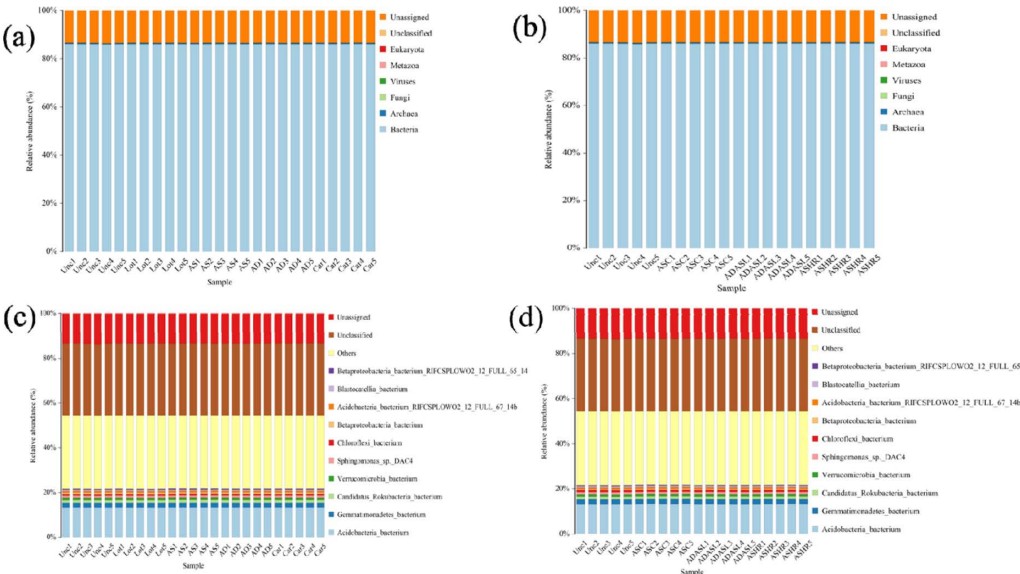

**Fig 4. Detection of species classification.** (a) and distribution of main species **(b)**, classification of bacteria (c) and detection of major bacterial distributions **(d)**.

roles in soil ecosystems. *Gemmatimonadetes* are commonly associated with drought tolerance and may contribute to microbial resilience in degraded soils. Their ability to withstand low-moisture conditions suggests that they may play a role in maintaining microbial activity and functionality under drought stress. *Verrucomicrobia*, on the other hand, have been linked to carbohydrate metabolism and organic matter decomposition, which are crucial for soil restoration. Their enzymatic capabilities enable them to degrade complex organic compounds, contributing to nutrient recycling and improving soil fertility. The presence of these bacterial groups highlights the complex interactions within microbial communities and their collective influence on soil restoration and stability. The composition of the microbial community is dynamic, and it can change in response to changes in environmental factors. For example, the impact of temperature increases on microbial communities is very large. Rising temperatures can promote the growth and reproduction of microbes, but may also lead to the death or decline of other types of microbes. Similarly, changes in land use patterns also have a greater impact on the composition of microbial communities. For example, changes in agricultural land may lead to significant reductions in certain components of the microbiome, thus affecting the stability of the ecosystem. In addition to these factors, changes in microbial communities are also influenced by several other factors, such as acidity, salinity, REDOX potential, and microbial activity.

### 3.3. Functional changes of microbial communities under different planting patterns

Functional annotations were performed on the acquired biological information, including general database annotations, such as KEGG, eggNOG, Pfam, SwissProt, NR, and GO, and special database annotations, such as CARD and CAZy [19]. The above-mentioned techniques were used to analyze functional genes under various planting patterns. The biogeochemical cycle of carbon/nitrogen has a profound significance for soil microecology. Our results show that the soil under the composite recovery model has more abundant carbon/nitrogen cycling functional genes, indicating the excellent characteristics of the composite model.

GO annotation revealed three major components: cellular, molecular, and biological. The high expression process corresponding to each component included the membrane reaction process (corresponding to cell activity) between the

cells and the catalytic activity of the molecular process (related to intracellular and extracellular enzymatic processes) and metabolic processes (Fig 5a). The results showed that the detection of functional genes was based on basic membrane transport, enzyme catalysis, and metabolic activities related to the life activities of biological cells, similar to the results of soil enzyme activity determination. Furthermore, we performed a statistical analysis of the functional genes related to the KEGG metabolic pathway at the secondary level (Fig 5b). The results showed that the detection of metabolism-related functional genes was mainly based on amino acid, carbohydrate, and energy metabolisms. This was consistent with the relative detection of proteases and hydrolases in the soil samples. Subsequently, the eggNOG functional gene function classification showed that amino acid transfer and metabolism, energy production and consumption, and carbohydrate transport and metabolism were the main gene functions (Fig 5c). The dominance of carbohydrate metabolism genes suggests enhanced microbial degradation of plant residues, facilitating soil organic matter accumulation. Similarly, increased amino acid metabolism may indicate active nitrogen cycling, which is critical for restoring degraded soils. Furthermore, the statistical analysis of the distribution of carbohydrate enzymes showed that the main carbohydrate enzymes were glycoside hydrolases, carbon-based transfer enzymes, polysaccharide lyases, carbohydrate esterases, and carbohydrate-binding modules (Fig 5d). Therefore, we detected several representative glycoside hydrolases; these are discussed in the first section of the results. Finally, we calculated the abundances of resistance genes in the soil samples (Fig 5e). The main components detected included aminoglycosides, tetracyclines, macrolides, peptides, and phenicols.

Microorganisms are the drivers of soil material transformation, and the higher the microbial diversity, the stronger the ecosystem service function is generally considered. Higher microbial diversity in mixed planting systems aligns with increased functional gene richness, suggesting that plant diversity promotes functionally diverse microbial communities. However, the relationship between microbial diversity and soil carbon and nitrogen transformation and utilization efficiency is unclear, which seriously affects the protection of high-quality soil resources and the improvement of medium and

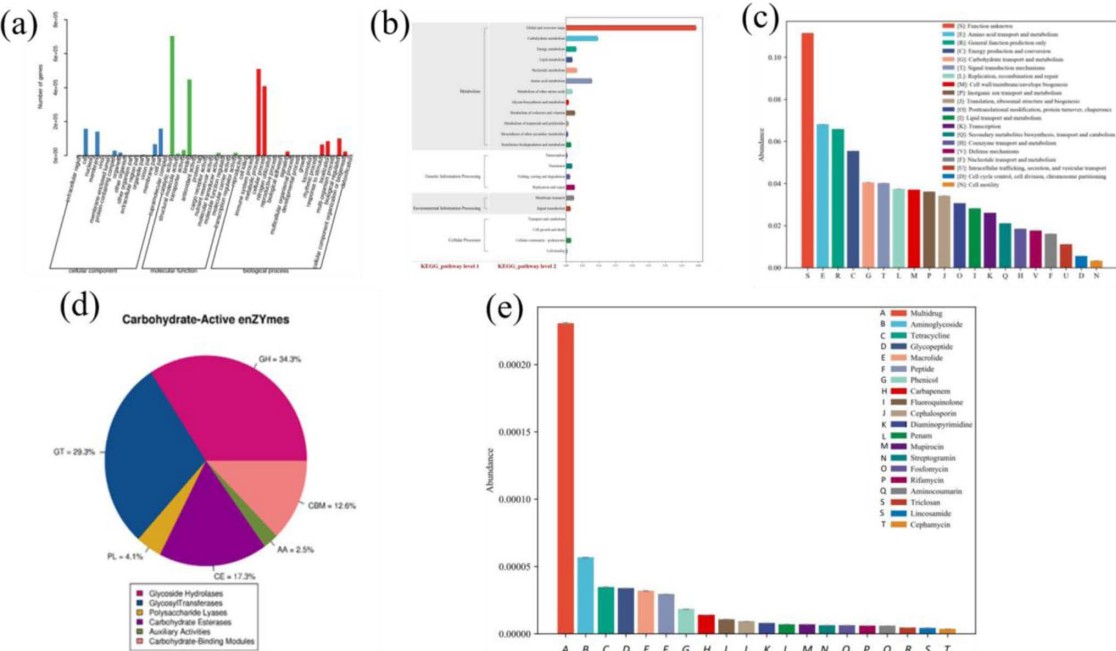

**Fig 5. Functional annotation analysis.** a, GO second-level node annotation classification statistics; b, KEGG metabolic pathway related functional genes at the second level; c, eggNOG functional gene functional classification statistics; d, carbohydrate enzyme distribution ratio diagram; e, antibiotics Resistance gene abundance statistics chart.

low-yield fields. The biogeochemical cycle mediated by the microbial community plays an important role in global nutrient (C and N) cycles and is sensitive to environmental changes [36]. Limited information on the composition and functional diversity of microorganisms that control biogeochemical cycles in environments with different planting patterns is available. Metagenomic analysis showed that a wide range of carbohydrate and energy metabolism genes were present in the declining soils. An in-depth study of the metagenomic data showed that protease and hydrolase genes were highly abundant, and important carbohydrate enzyme functional genes were detected as glycoside hydrolases, carbon group transferases, polysaccharide lyases, carbohydrate esterases, and carbohydrate-binding modules. This study showed that functional genes involved in carbohydrate anabolism, catabolism, and $CO_2$ fixation are widespread in the microbiome of degraded environments and have broad potential for the ecological restoration of production and characterization of ecologically degraded land [37]. The detection of functional genes for carbohydrate, protein, and energy metabolism indicated that material circulation and energy flow play important roles in ecological decline. Microbial communities are highly responsive and can respond in many different ways to changes in the environment. For example, when the temperature rises, the microbial community will adapt to the new environment and adjust, and this adaptation is achieved by changing the structure, number, and distribution of the microbial community to meet the requirements of the new environment. Additionally, environmental changes can also make certain microorganisms more prominent in the community. For example, some microorganisms adapted to harsh environments such as high temperatures and high salt may become more and more numerous when these environments change, and even become the dominant species in the community.

### 3.4. Diversity changes of microbial communities under different planting patterns

#### 3.4.1. β diversity analysis of species composition.
The β diversity analysis of species composition under different planting types was carried out through PCoA analysis, NMDS analysis, UPGMA analysis, and permanova analysis of principal components. The PCoA found that the structures of bacterial, viral, archaeal, and fungal communities differed significantly among planting types. Permanova analysis (permutational multivariate analysis of variance, also known as Adonis analysis) showed that the planting patterns affected the structure of the bacterial, archaeal, and fungal communities simultaneously (Fig 6). The results showed that there was little difference within the groups. However, there were significant differences between the groups, indicating that different planting types had significant effects on soil microbial community composition. This is because different plants tend to form different rhizosphere microorganisms, thus affecting the microorganisms in the surrounding soil owing to differences in the enrichment and release of enzymes and metabolic substrates by plants [38]. The mixed model promotes biodiversity and ensures the stability of microbial community structure. Mixed plant species create varied rhizosphere environments that promote microbial niche differentiation. This reduces competition and supports more diverse microbial interactions, ultimately leading to ecosystem stability. Different plant species exude different root exudates, including organic acids, sugars, amino acids, and secondary metabolites, which selectively enrich specific microbial populations. This process enhances functional complementarity among microbial taxa, reducing competitive exclusion and allowing a greater variety of microbes to coexist. Moreover, a diverse plant community increases soil structural complexity, providing microhabitats that further contribute to microbial diversity and resilience. In addition, forest restoration of degraded land can increase soil C concentration due to the input of accumulated organic matter (OM) from aboveground litter and root turnover [39]. Guo et al. found that soil C reserves increased by 53% when crops were converted into secondary forests [40]. The composition and content of C, N, P, and other nutrient elements in the soil are closely related to the composition and structure of the microbial community. Differences in the plant species used to restore vegetation cover led to the formation of completely different microbial communities in the soil. However, it is necessary to determine the ideal plant collocation patterns to improve the diversity and stability of soil microbial communities. The research has shown that multi-plant restoration of vegetation cover can result in higher and more abundant soil carbon and soil nitrogen species. Therefore, it is necessary to investigate soil microbial diversity under different cropping patterns.

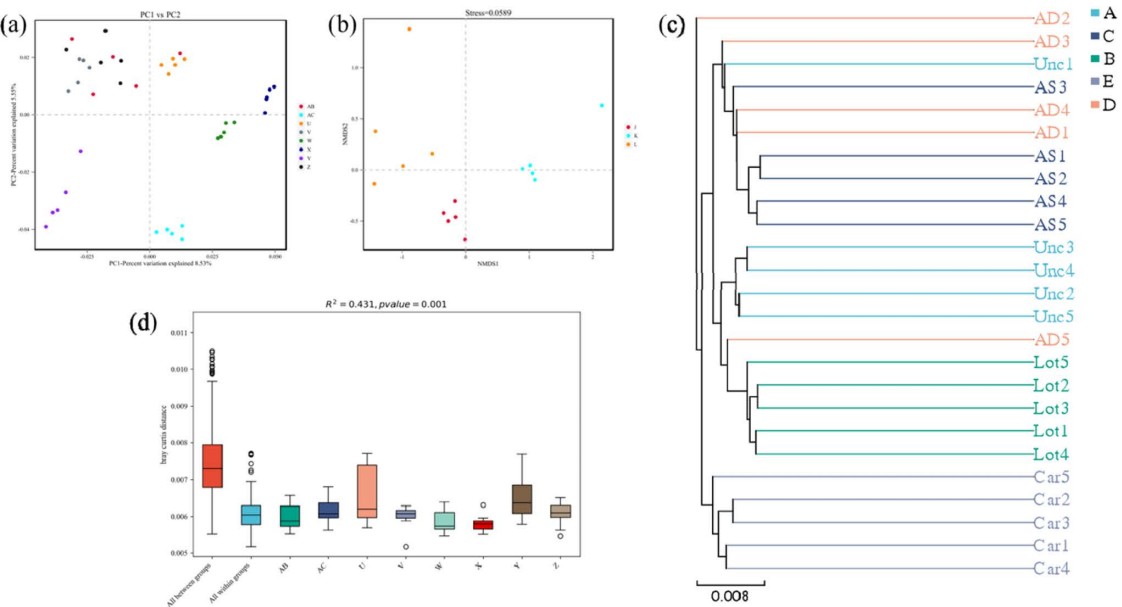

**Fig 6. β diversity analysis of species composition under various planting patterns.** a, PCoA analysis of principal components; b, NMDS analysis; c, UPGMA analysis; d, permanova analysis.

**3.4.2. Parameter test for differences in species composition between groups.** The rank-sum test is a non-parametric method. Among them, "rank" is also called rank; that is, the sum of the above sequence numbers is called "rank sum," and the rank sum test is a method of hypothesis testing using the rank sum as a statistic. In this study, the sampling rank-sum test was used to analyze differences in the composition of microbial communities under different planting types. Analyses of abandoned land versus single-planted forest and abandoned land versus mixed forest revealed that 61 and 58 species, respectively, were tested. The results showed that the classification tree of each group of samples showed significant differences among the groups, and the alfalfa land (lot) was similar to the abandoned land, with no significant differences among the forest lands. In general, bacterial abundance in the plantation area was significantly upregulated, among which the abundances of bacteria such as *actiobacteria*, *acidiobacillia, cytophagia*, *agaricomycetes*, *ardenticatenia*, and *thermoleophilia* were significantly upregulated (Fig 7a). A comparison of abandoned land and mixed forests revealed that the combination of mixed forests significantly affected the composition of the bacterial community (Fig 7b). In contrast to the single-planting forest area, the abundance of some bacteria in the mixed forest was downregulated, indicating that it was not the formation of dominant bacteria or the increase in intermediate competition. This indicates that compared to the monoculture forest, the microbial community formed in the mixed forest area tended to form the dominant species. The interspecific competition was less but more stable. This is because the mixed forest can provide soil with more abundant OM and rhizosphere secretions, increase the content of nutrient elements in the rhizosphere and non-rhizosphere soils, and increase the abundance of enzymes and substrates for microbial metabolism [41]. In addition, the mixing of tree species can alleviate the restriction of microbial utilization of nutrient elements in the soil [42]. Otherwise, micro-organisms play a variety of functions and roles in soil. First, they are involved in the decomposition and transformation of organic material, degrading organic material into a form that can be absorbed by plants and used to provide nutrients needed by crops. Secondly, microorganisms also participate in the formation of soil structure. Through the production of cellular adhesion substances and extracellular polysaccharides, they help to consolidate soil particles and form a good soil structure. In addition, microorganisms also form a symbiotic relationship with plant roots, jointly building rhizosphere ecosystems and promoting plant growth and immune ability. In

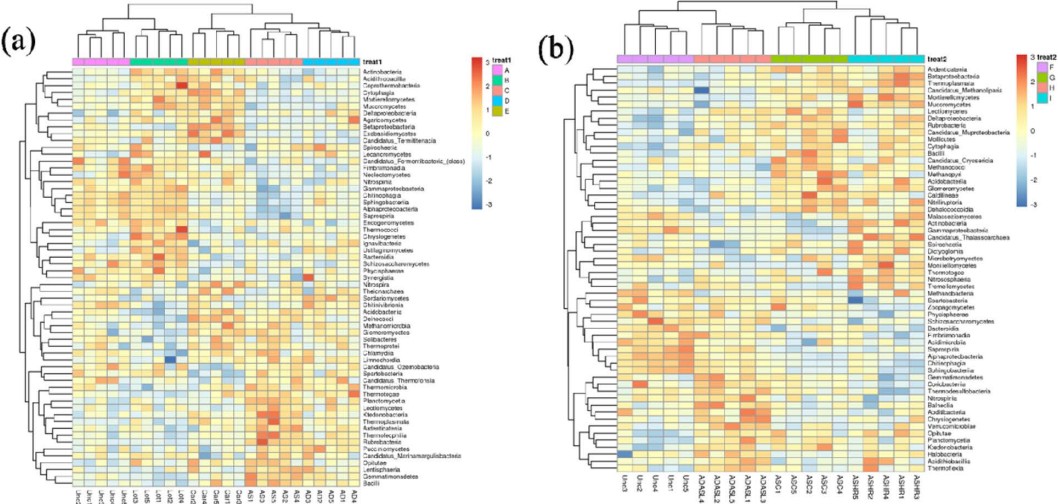

**Fig 7. Rank sum test difference species abundance heat map.** a, abandoned land vs. single planting area; b, abandoned land vs. mixed forest.

addition, microorganisms can also control the growth of pathogenic microorganisms in the soil and reduce the occurrence of diseases. Therefore, microbial diversity in farmland is closely related to soil quality. Higher levels of microbial diversity are generally associated with better soil quality, promoting soil nutrient cycling, improving crop yield and quality, and enhancing soil resistance and stability.

MetagenomeSeq is an R package used to test for differentially abundant genes and species. This software applies a new normalization method to avoid the effect of uneven sequencing depths [43]. It also uses a zero-expansion Gaussian distribution mixture model to solve the influence of abundance differences caused by insufficient sampling during inspection. In this study, the species shown were those with $p < 0.05$ after metagenomeSeq analysis. To this end, we selected a single planting area and a mixed forest area and used metagenomeSeq for further analysis. The results showed that compared to the abandoned land, the abundance of *Chitinivibrionia* in alfalfa land increased, while that of *Lecanoromycetes*, *Chrysiogenetes,* and *Ustilaginomycetes* was downregulated. Relatively inferior species with reduced abundances were detected in the mixed forests (Fig 8). Line discriminant analysis (LDA) effect size can identify biomarkers with statistical differences between different groups. Random forest analysis, a machine learning algorithm, is a classifier containing multiple decision trees that can efficiently and quickly select the most important species category (biomarker) based on the original sample classification. The abscissa, used for species importance ranking, is the species importance measurement; the ordinate is the species name sorted by importance (Fig 9). The species importance ranking results showed that *mucoromycota*, *proteobacteria*, *actinobacteria*, and *rokubacteria* were the four most important potential strains that played a crucial role in different soil use patterns in ecologically degraded areas. Microorganisms are key components of soil ecosystems and are essential for maintaining soil health. Microorganisms are involved in regulating soil pH value, improving soil structure, resisting environmental stress, and providing plant nutrition. Higher levels of microbial diversity contribute to the ecological stability and function of soils and maintain the health of farmland. Higher microbial diversity is linked to improved soil resilience, as diverse communities provide redundancy in ecological functions, ensuring stability under environmental stress. This redundancy allows the soil microbiome to maintain critical functions such as organic matter decomposition, nutrient cycling, and disease suppression even under adverse conditions. For instance, in degraded soils, a diverse microbial community can mitigate nutrient limitations by facilitating nutrient mobilization and enhancing plant-microbe interactions. Additionally, microbial diversity can improve soil aggregation, which enhances water retention

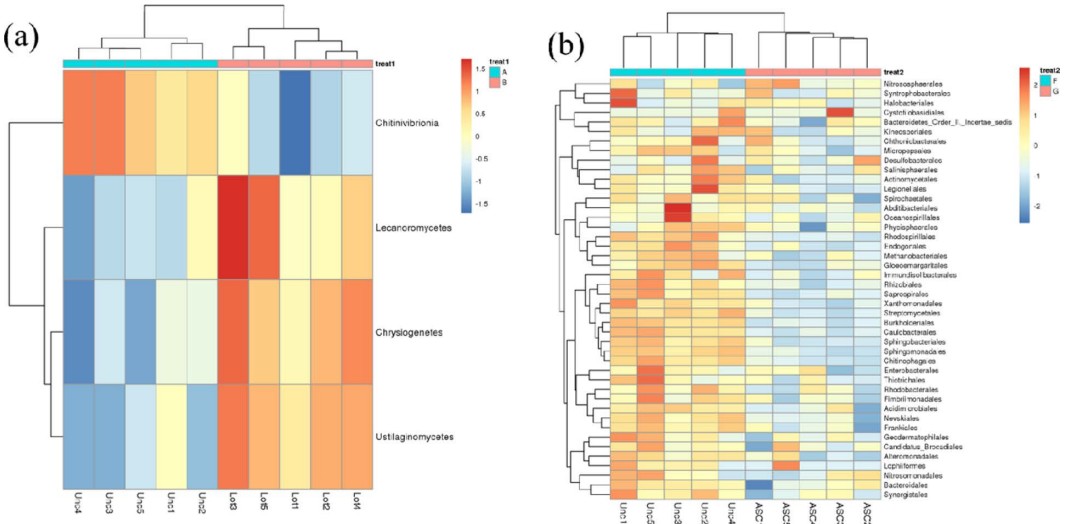

**Fig 8. MetagenomeSeq differential species abundance heat map.** a, abandoned land vs. single planting area; b, abandoned land vs. mixed forest.

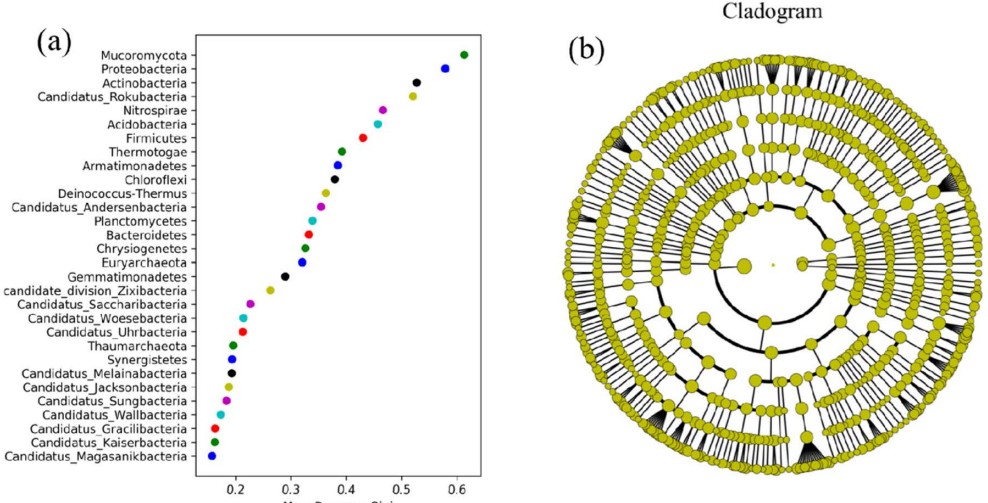

**Fig 9. Species importance analysis.** a, random forest analysis of species importance ranking diagram; b, lefse analysis of evolutionary branch diagram.

and reduces soil erosion, further promoting ecosystem stability. Understanding microbial succession under different planting patterns can inform sustainable land management strategies, particularly in degraded ecosystems. By studying how microbial communities respond to various cropping systems, it is possible to design agricultural and reforestation practices that enhance soil biodiversity and resilience. Long-term shifts in microbial composition influence nutrient cycling, organic matter accumulation, and the suppression of soilborne pathogens, all of which are essential for ecosystem restoration and productivity. Therefore, integrating microbial community dynamics into land management plans can provide a foundation for more sustainable and adaptive agricultural systems.

### 3.4.3. Effects of different planting patterns on the structure of soil microbial communities.

Soil microecosystems are the largest and most complex ecosystems worldwide. Microorganisms account for the largest proportion of the world's biodiversity, play a key role in the biogeochemical cycle, and are the basic driving factors that determine the primary productivity of soil [44]. However, owing to the participation of various human activities, the composition and structure of the soil microecological microbial community may be disturbed. Planting type was found to have a significant effect on the structure and composition of indigenous microbial communities. The soil microbial composition and structure in the mixed forest were more stable than those in the single planting area. Over the long term, the transition from degraded soil ecology to artificially managed restorative soil ecology significantly affects soil carbon and nutrient content, texture, and pH [45,46]. Changes in soil physical and chemical factors may stem from changes in soil bacterial diversity and function [47]. Therefore, the diversity and structural stability of soil bacterial communities are key factors in understanding the impact of planting pattern changes on agricultural ecosystems and human health. Herein, we found that in areas where land was declining, the type of planting profoundly affected the composition and structure of soil microbial communities, and mixed planting of multiple crops was beneficial to the internal stability of the soil microecology. The dominant rhizosphere microorganisms formed by different plant roots were different, and the resistance of a single microbial composition to harmful environmental factors was limited. In the composite model, the complex microbial composition ensured the complexity and heterogeneity of soil microbial community structure and enhanced the resistance of soil microorganisms.

Micro-organisms are important nutrient converters in soil. Different kinds of microorganisms participate in the decomposition and transformation process of organic matter in soil, decomposing organic matter into inorganic nutrients that can be absorbed and utilized by plants, such as nitrogen, phosphorus, potassium, etc. Microorganisms release nutrients by breaking down organic matter and releasing these nutrients into the soil to feed plant growth and development. Therefore, higher levels of microbial diversity are generally associated with better nutrient cycling and soil quality.

## 3.5. Species correlation analysis

A correlation network diagram is a form of correlation analysis. Eighty species with the highest abundance were screened. The 54 identifiable species were selected for subsequent analysis. According to the abundance and change of each species in each sample, the Spearman algorithm was used for correlation analysis (including positive correlation and negative correlation), and a statistical test was performed using a data set with a correlation > 0.5 and a $p < 0.05$, and a correlation network diagram was drawn based on Python. The results showed that the expression of C- and N-cycle-related bacteria was closely related (Fig 10), indicating that the species cycle still plays an important role in the restoration of degraded soil ecology in different restorative planting areas [48]. Moreover, microorganisms promote the agglomeration of soil particles by producing adhesive substances and exopolysaccharides to form a good soil structure. Good soil structure helps water infiltration and retention, provides water and oxygen needed by plants, and improves soil permeability and water retention. Through the activity of the cell body and the secretion of metabolites, microorganisms promote the combination of soil particles and organic matter to form stable soil aggregates and improve soil quality.

## 4. Conclusion

Microbial community composition and dynamics are shaped by various environmental factors, including temperature, moisture, oxygen levels, nutrient availability, and land use. These communities play a vital role in ecosystem stability, necessitating sustainable land management practices to preserve soil microbial diversity.

This study demonstrated that topography and planting patterns significantly influence soil microbial composition and function. Mixed forests had the highest soil carbon (C) and nitrogen (N) content, along with the greatest microbial abundance and diversity. In contrast, single-species planting systems showed lower microbial diversity, with fewer dominant species, making them more susceptible to biodiversity loss. The increased microbial abundance in mixed

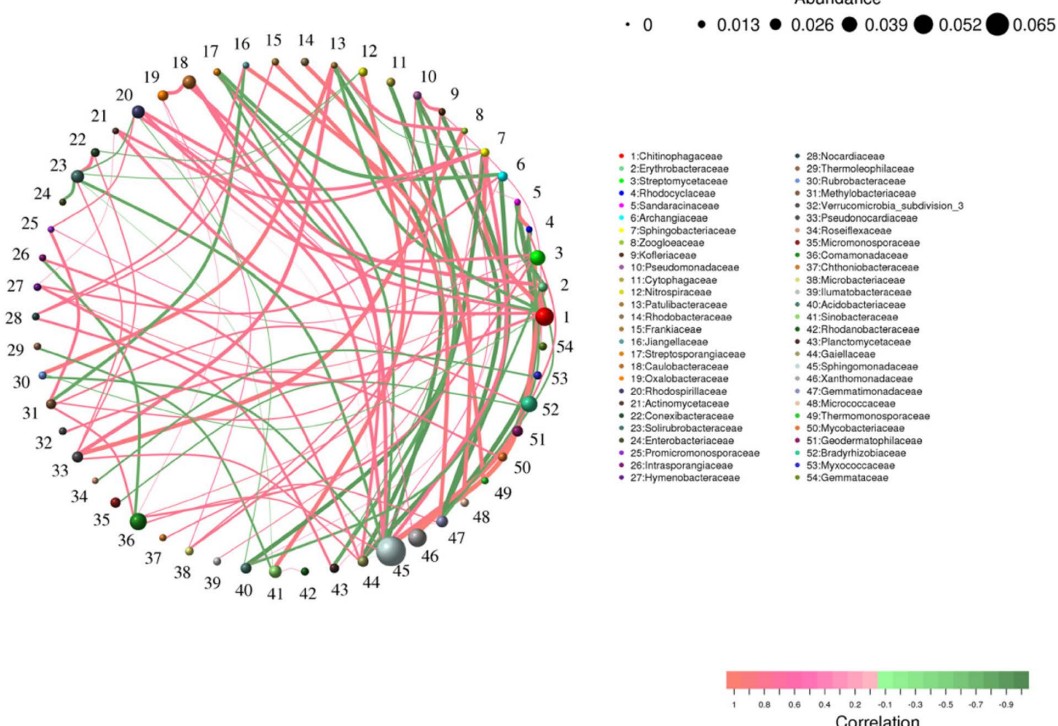

**Fig 10. Neural network analysis diagram: the circle represents the species, the size of the circle represents the abundance; the line represents the correlation between the two species, the thickness of the line represents the strength of the correlation, the color of the line: orange represents positive correlation, and green represents negative correlation.**

forests may be attributed to a more diverse range of root exudates, which provide varied carbon sources that support a broader microbial community. Erosion and biodiversity loss were more pronounced in single-planting areas, where large soil aggregates had lower carbon density. Microbial abundance in mixed forests was 32.1% higher than in single-species plots, likely due to the more diverse rhizosphere environment supporting a broader range of microorganisms. Functional gene analysis revealed that energy metabolism and species interactions are key to stabilizing degraded ecosystems. Proper management of planting patterns, particularly through mixed forests, can enhance microbial diversity and ecological resilience. These findings provide valuable insights for soil restoration efforts in fragile ecosystems. While this study provides valuable data support for future soil remediation in ecologically fragile areas, it is limited by its short-term analysis. Future work should explore the long-term stability of microbial communities under different planting systems and investigate the interactions between plant root exudates and microbial metabolic pathways, while assessing seasonal and long-term microbial dynamics to fully understand the sustainability of mixed-planting systems.

## Author contributions

**Data curation:** Gang Chen, Xia Wu, Yangyang Bai, Yan Wu.

**Investigation:** Yitong Liu, Tianning Wang.

**Validation:** Jinjun Cai.

**Writing – original draft:** Weiqian Li.

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
