## [Decision Letter · Decision Letter 0]

22 Sep 2024

PONE-D-24-15257Microbial community succession mediated by planting patterns in the loess beam-shaped hilly region of ChinaPLOS ONE

Dear Dr. Cai,

Thank you for submitting your manuscript to PLOS ONE. After careful consideration, we feel that it has merit but does not fully meet PLOS ONE’s publication criteria as it currently stands. Therefore, we invite you to submit a revised version of the manuscript that addresses the points raised during the review process. Please submit your revised manuscript by Nov 06 2024 11:59PM. If you will need more time than this to complete your revisions, please reply to this message or contact the journal office at plosone@plos.org . Please include the following items when submitting your revised manuscript:

We look forward to receiving your revised manuscript.

Kind regards,

Massimiliano Cardinale, PhD

Academic Editor

PLOS ONE

“The authors wish to express their sincere thanks to the Ningxia Hui Autonomous Region Natural Science Foundation�2022AAC03453�, and the Ningxia Hui Autonomous Region Key R&D Project(2023BEG02042). In addition, this work also is supported by the special project of Ningxia Academy of Agriculture and Forestry Science with foreign (DW-X-2020002), the Ningxia Hui Autonomous Region Sci-Tech innovation demonstration program of High-quality agricultural development and ecological conservation (NGSB-2021-11-01). The authors thank Biomarker Technologies for the support of tests.”

“The authors wish to express their sincere thanks to the Ningxia Hui Autonomous Region Natural Science Foundation�2022AAC03453�, and the Ningxia Hui Autonomous Region Key R&D Project(2023BEG02042). In addition, this work also is supported by the special project of Ningxia Academy of Agriculture and Forestry Science with foreign (DW-X-2020002), the Ningxia Hui Autonomous Region Sci-Tech innovation demonstration program of High-quality agricultural development and ecological conservation (NGSB-2021-11-01). The authors thank Biomarker Technologies for the support of tests.”

“no”

Reviewers' comments:

Reviewer's Responses to Questions

**Comments to the Author**

1. Is the manuscript technically sound, and do the data support the conclusions?

Reviewer #1: Yes

Reviewer #2: Yes

2. Has the statistical analysis been performed appropriately and rigorously? 

Reviewer #1: Yes

Reviewer #2: Yes

3. Have the authors made all data underlying the findings in their manuscript fully available?

Reviewer #1: Yes

Reviewer #2: Yes

4. Is the manuscript presented in an intelligible fashion and written in standard English?

Reviewer #1: Yes

Reviewer #2: Yes

5. Review Comments to the Author

Reviewer #1: The manuscript submitted to the journal titled “Microbial community succession mediated by planting patterns in the loess beam-shaped hilly region of China”. The succession of microbial community structure and function in ecologically fragile areas is a core ecological issue for solving the problem of ecological degradation. This research focuses on the functional dynamics of microbial communities in soil sample sequences of different planting areas in ecologically degraded areas. It is important and contributes with information about the succession of microorganisms in the loess beam-shaped hilly region of China. However, there are some suggestions as follows:

1. About the co-author, why “Tianning WANG” NOT “Tianning Wang”

2. In abstract, you mentioned “we studied the functional dynamics……” but I can not find any “dynamics”.

3. Lack of lots of necessary data in abstract, such as in Line 19-21, “Bacteria showed higher succession variability than fungi, and changes in bacterial and fungal communities were related to soil vegetation type and physical, chemical, and enzyme activity characteristics.” So, what physical, chemical, and enzyme activity parameter�and how these parameters influence microbial community?

4. Line 80, Change "Therefor" for "In this study".

5. Line 90, Change “Materials and methods” for “Materials and Methods”.

6. In “2.1 Study area and soil sampling”, the author introduced the research site, including latitude and longitude. Please add a figure to show this site will be better.

7. In “2.2 Soil sampling……” has “soil sampling” repeated with 2.1, so delete “soil sampling” in 2.1.

8. Line 108-109 what “+” in “forest +” “shrub +”means?

9. Line 127, Please confirm the wavelength in the molybdenum blue method? “700 nm” or “880 nm”?

10. Line 134, change “2.4.1 Extraction of DNA from soil bacteria and fungi” for “2.4.1 Extraction of DNA from soil samples”

11. Why do you did “qRT-PCR quantitative analysis”?

12. Line 144, what is “H O “ mean? Maybe the author want to write “H2O”?

13. There is so much blank between Line 175 and Line 176.

14. Line 176, change “3. Analysis of results” for “3. Results and Discussion”.

15. Line 176-183. These two sentences belong to “Materials and Methods”, not belong here.

16. The title of table 2 is not proper. Suggest change to “soil physical and chemical characteristics”.

17. Line 212-214, plus bacteria and fungi, archaea, and virus is not 100%?

18. Table 3 is not necessary for results, put it in the “Supplement Materials”.

19. From the Figure 3, it seems it is the same of all the samples, has no difference. How to interpret?

20. The font size seems strange. It is bigger of the title of figures compare to main body.

21. In Figure 5, why (c) use red? And what are “A,B,C,D,E” mean? It not clear.

22. In Figure 9, this network figure is a little disordered? What is the main factor?

23. The figures quality should be improved especially Figure 3, Figure 4, Figure 6, Figure 7 and so on.

24. Line 407, Line 442, Line 481, What is the 7th, 21th, 36th reference going on? And the font size in references are different, please check one by one.

Reviewer #2: Review Report has been added as an attachment since it exceeds 20000 characters. I recommend major revisions to address clarity, enhance ecological relevance, and expand on the practical applications of the research. With these improvements, the manuscript has strong potential for publication.

6. PLOS authors have the option to publish the peer review history of their article (what does this mean? ). If published, this will include your full peer review and any attached files.

**Do you want your identity to be public for this peer review?** For information about this choice, including consent withdrawal, please see our Privacy Policy .

Reviewer #1: No

Reviewer #2: **Yes: ** BRIGHT FAFALI DOGBEY

---

## [Author Response · Author response to Decision Letter 1]

19 Jan 2025

We are grateful to the constructive comments and valuable suggestions made by the editorial board and the two reviewers, which have helped us to improve the quality of this paper. We have addressed all the comments and suggestions in the revised manuscript. Pleases see the detailed point-to-point responses to the comments in “Responses to Reviewers”.

---

## [Decision Letter · Decision Letter 1]

19 Feb 2025

PONE-D-24-15257R1Microbial Community Succession Mediated by Planting Patterns in the Loess Plateau, China: Implications for Ecological RestorationPLOS ONE

Dear Dr. Cai,

Thank you for submitting your manuscript to PLOS ONE. After careful consideration, we feel that it has merit but does not fully meet PLOS ONE’s publication criteria as it currently stands. Therefore, we invite you to submit a revised version of the manuscript that addresses the points raised during the review process.

We look forward to receiving your revised manuscript.

Kind regards,

Massimiliano Cardinale, PhD

Academic Editor

PLOS ONE

**Journal Requirements:**

Reviewers' comments:

Reviewer's Responses to Questions

**Comments to the Author**

1. If the authors have adequately addressed your comments raised in a previous round of review and you feel that this manuscript is now acceptable for publication, you may indicate that here to bypass the “Comments to the Author” section, enter your conflict of interest statement in the “Confidential to Editor” section, and submit your "Accept" recommendation.

Reviewer #1: All comments have been addressed

Reviewer #2: All comments have been addressed

2. Is the manuscript technically sound, and do the data support the conclusions?

Reviewer #1: Yes

Reviewer #2: Yes

3. Has the statistical analysis been performed appropriately and rigorously? 

Reviewer #1: Yes

Reviewer #2: (No Response)

4. Have the authors made all data underlying the findings in their manuscript fully available?

Reviewer #1: Yes

Reviewer #2: (No Response)

5. Is the manuscript presented in an intelligible fashion and written in standard English?

Reviewer #1: Yes

Reviewer #2: Yes

6. Review Comments to the Author

**Reviewer #1:**  (No Response)

**Reviewer #2: ** Thank you for your efforts in revising the manuscript. Your study presents important insights into the role of mixed planting systems in shaping soil microbial diversity and stability. The revisions have improved the clarity and depth of the manuscript; however, there are still areas that require further refinement. Below are my detailed comments to help enhance the manuscript before publication.

7. PLOS authors have the option to publish the peer review history of their article (what does this mean? ). If published, this will include your full peer review and any attached files.

**Do you want your identity to be public for this peer review?** For information about this choice, including consent withdrawal, please see our Privacy Policy .

Reviewer #1: No

Reviewer #2: **Yes: ** Bright Fafali Dogbey

---

## [Author Response · Author response to Decision Letter 2]

15 Apr 2025

Thanks editor and the two reviewers for the very constructive comments. We have revised our manuscript carefully following the comments and provide specific responses to each of the reviewers’ comments in blue. To make it easier for you to see where changes have been made in the manuscript, all major changes are in red.

---

## [Decision Letter · Decision Letter 2]

2 May 2025

Microbial Community Succession Mediated by Planting Patterns in the Loess Plateau, China: Implications for Ecological Restoration

PONE-D-24-15257R2

Dear Dr. Cai,

We’re pleased to inform you that your manuscript has been judged scientifically suitable for publication and will be formally accepted for publication once it meets all outstanding technical requirements.

Kind regards,

Massimiliano Cardinale, PhD

Academic Editor

PLOS ONE

Additional Editor Comments (optional):

Reviewers' comments:

Reviewer's Responses to Questions

**Comments to the Author**

1. If the authors have adequately addressed your comments raised in a previous round of review and you feel that this manuscript is now acceptable for publication, you may indicate that here to bypass the “Comments to the Author” section, enter your conflict of interest statement in the “Confidential to Editor” section, and submit your "Accept" recommendation.

Reviewer #2: All comments have been addressed

2. Is the manuscript technically sound, and do the data support the conclusions?

Reviewer #2: Yes

3. Has the statistical analysis been performed appropriately and rigorously? 

Reviewer #2: Yes

4. Have the authors made all data underlying the findings in their manuscript fully available?

Reviewer #2: Yes

5. Is the manuscript presented in an intelligible fashion and written in standard English?

Reviewer #2: Yes

6. Review Comments to the Author

Reviewer #2: The review report has been attached. Please refer to the document named Microbial Community Succession Revision Review Report

7. PLOS authors have the option to publish the peer review history of their article (what does this mean? ). If published, this will include your full peer review and any attached files.

**Do you want your identity to be public for this peer review?** For information about this choice, including consent withdrawal, please see our Privacy Policy .

Reviewer #2: **Yes: ** Bright Fafali Dogbey

---

## [Editor Report · Acceptance letter]

PONE-D-24-15257R2

PLOS ONE

Dear Dr. Cai,

I'm pleased to inform you that your manuscript has been deemed suitable for publication in PLOS ONE. Congratulations! Your manuscript is now being handed over to our production team.

Kind regards,

on behalf of

Dr. Massimiliano Cardinale

Academic Editor

PLOS ONE